# SpecMAS: A Multi-Agent System for Self-Verifying System Generation via Formal Model Checking

**Rishabh Agrawal[1]\*  Kaushik Tushar Ranade[1]\*  Aja Khanal[1]**
**Kalyan S. Basu[2]   Apurva Narayan[1]**
[1]Department of Computer Science, University of Western Ontario, London, Ontario
[2]ICASSSD, Cambridge, Ontario
{ragrawa9,kranade,akhanal3,apurva.narayan}@uwo.ca   ks.basu@gmail.com

## Abstract

We present SpecMAS, a novel multi-agent system that autonomously constructs and formally verifies executable system models from natural language specifications. Given a Standard Operating Procedure (SOP) describing a target system, SpecMAS parses the specification, identifies relevant operational modes, variables, transitions, and properties, and generates a formal model in NuSMV code syntax, an industry-standard symbolic model checker. A dedicated reasoning agent extracts both explicit and implicit properties from the SOP, and verification is performed via temporal logic model checking. If any properties fail to verify, an autonomous debugging agent analyzes counterexamples and iteratively corrects the model until all properties are satisfied. This closed-loop system design guarantees provable correctness by construction and advances the state of the art in automated, interpretable, and deployable verification pipelines. We demonstrate the generality, correctness, and practical feasibility of SpecMAS across a set of representative case studies and propose a new benchmark dataset for the evaluation and comparison of model checking performance. We release our code and benchmark dataset github.com/Idsl-group/SpecMAS.

## 1   Introduction

The increasing complexity and autonomy of modern software systems have made it critical to develop frameworks that not only generate such systems from specifications but also provide formal guarantees about their behavior. Formal specifications define pre-/post-conditions, invariants, and other assertions that must hold throughout system execution [1, 2]. In mission-critical domains such as healthcare, aviation, and industrial automation, informal reasoning and empirical testing are not enough [3, 4]. This is especially true when AI agents are use to generate and execute the system. In such cases, a proper testing suite or expert-defined system properties may not be available to verify correctness of the system while it is being generated [5]. This necessitates approaches that integrate both system synthesis and formal verification in a closed-loop manner.

In this paper, we propose SpecMAS, a novel multi-agent framework that autonomously ingests system specifications in the form of a Standard Operating Procedure (SOP) and constructs a complete formal system model using temporal logic and symbolic representations (via NuSMV [6]). We extract system states, transitions, and property specifications, and systematically verify their correctness using model checking [7]. To ensure comprehensive verification, our framework includes a dedicated reasoning agent [8] that identifies and encodes all properties. Explicit properties are directly extracted from the SOP, while implicit ones are inferred through a property exploration process driven by semantic analysis and contextual reasoning. If any specification properties fail during verification,

---

\*Equal Contribution

39th Conference on Neural Information Processing Systems (NeurIPS 2025).

our system leverages a dedicated Model Checker Debugging Agent that interprets counterexample traces to iteratively revise and correct the system model or properties until all generated specifications pass verification checks.

## 1.1 Objectives and Contributions

1. **Automated Model Synthesis**: Design a multi-agent system capable of extracting operational modes, variables, and transitions from natural language SOPs and translating them into formal NuSMV models.

2. **End-to-End Formal Verification**: Use symbolic model checking to ensure that all temporal logic properties (e.g., invariants, safety, reachability) are satisfied by the synthesized model.

3. **Recursive Self-Correction**: Introduce an agentic feedback loop that interprets NuSMV counterexamples and autonomously revises faulty specifications or transitions until full verification is achieved.

4. **Theoretical Guarantees**: Provide a foundation for building agentic systems that are correct by construction, thus bridging AI generated systems with formal logic and safety.

5. **Scalability and Generality**: Demonstrate the generalisability of SpecMAS across diverse domains through empirical validation and analysis on MiniSpecBench dataset with both synthetic and expert-designed formal models.

A key contribution of SpecMAS is its direct use of NuSMV, an industry-standard tool for symbolic model checking. Unlike prior work that relies on custom languages or proprietary compilers, our framework generates models in NuSMV's native specification language, enabling seamless integration with existing verification toolchains. This design choice reduces adoption barriers, eliminates the need for custom infrastructure, and leverages a mature ecosystem with robust support for temporal logic, counterexample generation, and scalability. By merging model synthesis, formal verification, and self-correction within a unified multi-agent framework, SpecMAS represents a step toward trustworthy autonomous systems capable of reasoning about the correctness of their own design.

## 2 Previous Work

Model checking is a formal verification technique that systematically explores the state space of a system model to determine whether it satisfies specified properties, typically expressed in temporal logic such as Computation Tree Logic (CTL) or Linear Temporal Logic (LTL). It has been widely adopted in safety-critical domains due to its ability to provide exhaustive guarantees about system behavior under all possible execution paths. Tools like NuSMV have established themselves as industry standards by offering symbolic and explicit-state verification capabilities. However, despite its rigor, model checking is traditionally a manual process that requires expert knowledge to construct accurate models and formal specifications. Recent research has explored automating parts of the process but poses a significant barrier to scalability and real-world deployment.

### 2.1 Model Translation and DSL-Based Methods

Several model translation and domain-specific language (DSL) approaches have been proposed to facilitate formal verification. Preibusch and Kammüller [9] presented a framework for translating Object-Z specifications to SMV, preserving object-oriented behavior but requiring manual specification. Zhongsheng et al. [10] compared SPIN, UPPAAL, and NuSMV, finding NuSMV more suited for distributed systems due to its support for CTL and LTL. DSL-based methods like MoSt [7] and NL2CTL [11] improve model structure by translating annotated or natural language inputs into temporal logic, though they rely on structured inputs or synthetic datasets. Liu et al. [12] introduced a Property Specification Language (PSL) tailored to smart contracts, offering formal rule generation but with limited generalizability. While these methods enhance automation and clarity, they are often domain-specific and depend on controlled input formats or prior modeling expertise.

### 2.2 LLM-Assisted Specification Synthesis and Verification

Wen et al. [13] introduced AutoSpec, a hybrid framework combining static analysis and program verification with LLM-based specification generation. The system incrementally synthesizes pre-

Table 1: Addressing limitations of prior work with the proposed SpecMAS method.

| Limitation Category | SpecMAS Solution |
| --- | --- |
| **Fragmented Scope** | Integrates system **model generation**, **property extraction**, and **recursive verification** into a single multi-agent pipeline. |
| **Domain Lock-in** | Designed to be **domain-agnostic**, able to operate across SOPs from various fields (e.g., manufacturing, logistics, control systems). |
| **Post-hoc Verification** | Ensures **correctness-by-construction** by embedding verification directly into the synthesis loop. |
| **No Transition Model** | Explicitly constructs full **NuSMV transition systems** (states, variables, transitions) based on SOPs. |
| **Shallow Property Coverage** | Extracts both **explicit and implicit properties** using reasoning agents, covering invariants, liveness, and safety conditions. |
| **No Debugging or Correction** | Includes a **self-corrective debugging agent** that uses counterexamples to iteratively fix unverifiable models. |
| **Non-interpretable Outputs** | Outputs models in **NuSMV**, an industry-standard formal language, ensuring transparency and adoption feasibility. |

conditions, postconditions, and loop invariants for C programs and validates them using theorem provers. Similarly, Ma et al. [14] presented SpecGen, a two-phase approach for formal specification generation in Java Modeling Language using LLMs. The first phase engages the LLM in a conversational prompt-driven loop, refining outputs based on verifier feedback. When this fails, a second mutation-based mechanism generates specification variants, which are heuristically selected for verification. SpecGen outperformed traditional tools [15, 16] which utilize templates defined by human experts to generate candidate specifications. However, such models are fine-tuned for specific domains (e.g., C/Java programs, smart contracts), limiting their adaptability.

Loop invariants are critical for verifying programs with iterative behavior, yet generating them remains a long-standing challenge. Wu et al. [17] introduced a neuro-symbolic framework that leverages LLMs to generate candidate invariants in a "query-filter-reassemble" loop. Bounded Model Checking (BMC) [18] is used to filter invalid predicates, and surviving clauses are recombined into new invariant candidates. Pirzada et al. [19] proposed a novel extension of BMC that replaces loop unrolling with LLM-generated invariants, verified using a first-order theorem prover [20]. This method was able to prove correctness for programs that other tools [21, 22] failed to verify.

## 3 Proposed Method

We propose **SpecMAS**, a multi-agent system that automates the synthesis and verification of formal models from natural language SOP documents describing a system. At the core of our framework lies the formal verification of system behavior using *model checking*, grounded in the semantics of Kripke structures. Formally, we represent the synthesized system as a finite-state transition system (Kripke structure) defined by the tuple:

$$\mathcal{M} = \langle S, S_0, R, L \rangle \tag{1}$$

where, $S$ is a finite set of states, $S_0 \subseteq S$ is the set of initial states, $R \subseteq S \times S$ is a total transition relation capturing all valid system behaviors, and $L : S \to 2^{AP}$ is a labeling function mapping each state to a set of atomic propositions that hold in that state. Given a temporal logic specification $\varphi$, the verification task reduces to checking whether the system satisfies the property, denoted as:

$$\mathcal{M} \models \varphi. \tag{2}$$

SpecMAS constructs $\mathcal{M}$ directly from the SOP by identifying system states, variables, and transition conditions through a structured parsing and reasoning pipeline. Properties $\varphi$ are either explicitly extracted from the SOP or inferred through a property exploration agent. The model is then passed to a symbolic model checker (NuSMV), which determines the satisfaction of $\varphi$. LTL properties are

interpreted over paths; CTL combines path quantifiers $A$ (for all paths) and $E$ (there exists a path) with temporal operators $X$, $F$, $G$, $U$.

**Input:** Standard Operating Procedure document SOP
**Output:** Machine–verified NuSMV model $\mathcal{M}_{\mathtt{OK}}$

**Pipeline I – Kripke Model & Specification Synthesis**;
**begin**                                           // Phase 1 - Information Gathering
  | TaskInfo $\leftarrow$ EXTRACTTASKINFORMATION(SOP, KB);
**end**
**begin**                                           // Phase 2 - Kripke Model Generator
  | $\mathcal{K} \leftarrow$ KRIPKEEXTRACTOR(TaskInfo);
  | $\mathcal{K} \leftarrow$ KRIPKESOLVER($\mathcal{K}$, ReACT, KB);
  | $\mathcal{K} \leftarrow$ SELFCORRECT($\mathcal{K}$, ReACT);
  | $\mathcal{M}\_\mathtt{raw}$ $\leftarrow$ SERIALIZETONuSMV($\mathcal{K}$);
**end**
**begin**                                           // Phase 3 - Specification Generator
  | $\varphi \leftarrow$ PROPERTYEXTRACTOR(SOP, KB);
  | $\varphi \leftarrow$ PROPERTYSOLVER($\varphi$, ReACT);
  | Specs_raw $\leftarrow$ SERIALIZESPECS($\varphi$);
**end**

**Pipeline II – Model-Checking and Debugging**;
$\mathcal{M} \leftarrow \mathcal{M}\_\mathtt{raw}$,   Specs $\leftarrow$ Specs_raw;
RevHist $\leftarrow \{\}$                  // initial revision history;
**repeat**                                              // coding-debugging loop
  **if** STATUS $= \mathit{Verified}$ **then**
    | **return** $(\mathcal{M},$ Specs$)$                // all properties satisfied
  **else if** STATUS $= \mathit{SyntaxError}$ **then**
    | $(\mathcal{M}, \text{Specs}) \leftarrow$ FIXSYNTAX($\mathcal{M}, \text{Specs}, \text{Refine}, \text{KB}, \text{RevHist}$);
  **else if** STATUS $= \mathit{CounterExample}$ **then**
    | $(\mathcal{M}, \text{Specs}) \leftarrow$ GATHERCONTEXT&REFINE($\mathcal{M}, \text{Specs}, \text{Plan}, \text{KB}, \text{RevHist}$);
  **end**
  **else**                                             // unsupported feature
    | $(\mathcal{M}, \text{Specs}) \leftarrow$ FIXSEMANTICS($\mathcal{M}, \text{Specs}, \text{feedback}, \text{KB}, \text{RevHist}$);
  **end**
  RevHist $\leftarrow$ RevHist $\cup \{(\mathcal{M}, \text{Specs})\}$;
**until** (STATUS, feedback) $\leftarrow$ RUNNuSMV($\mathcal{M}, \mathit{Specs}$);

        **Algorithm 1:** SPECMAS– End-to-End Generation and Verification

In both Linear Temporal Logic (LTL) and Computation Tree Logic (CTL) model checking, a *counter-example* is a witness path (or tree) that demonstrates why a temporal-logic specification $\varphi$ is false for a Kripke structure $\mathcal{M} = \langle S, S_0, R, L \rangle$. For **LTL**, the violation is witnessed by an *infinite ultimately-periodic path*:

$$\pi = s_0 s_1 \ldots s_k \, (s_{k+1} \ldots s_{k+\ell})^\omega, \qquad s_0 \in S_0, \ (s_i, s_{i+1}) \in R, \tag{3}$$

whose finite prefix of length $k$ is followed by a loop of length $\ell > 0$; by construction $\pi \models_{\text{LTL}} \neg\varphi$. Because $\mathcal{M}$ is finite, every counter-example returned by a model checker can be represented in this $\alpha\beta^\omega$ form, where $\alpha = s_0 \ldots s_k$ and $\beta = s_{k+1} \ldots s_{k+\ell}$.

For **CTL**, a property failure is shown by a finite path:

$$\pi = s_0 s_1 \ldots s_m, \qquad s_0 \in S_0, \ (s_i, s_{i+1}) \in R, \tag{4}$$

or, for universal violations, a finite tree, that ends in a state $s_m$ violating the required sub-formula. Each position $i$ can be annotated with the set:

$$\mathcal{F}_i = \{\psi \subseteq \varphi \mid \mathcal{M}, s_i \not\models \psi\}, \tag{5}$$

so the complete counter-example is the pair $\big(\pi, \, (\mathcal{F}_0, \ldots, \mathcal{F}_m)\big)$ and the non-empty set $\mathcal{F}_m$ pinpoints exactly which obligations are violated. Thus, an LTL counter-example is mathematically an ultimately-periodic path $\alpha\beta^\omega$ with $\pi \models \neg\varphi$, whereas a CTL counter-example is a finite, transition-consistent

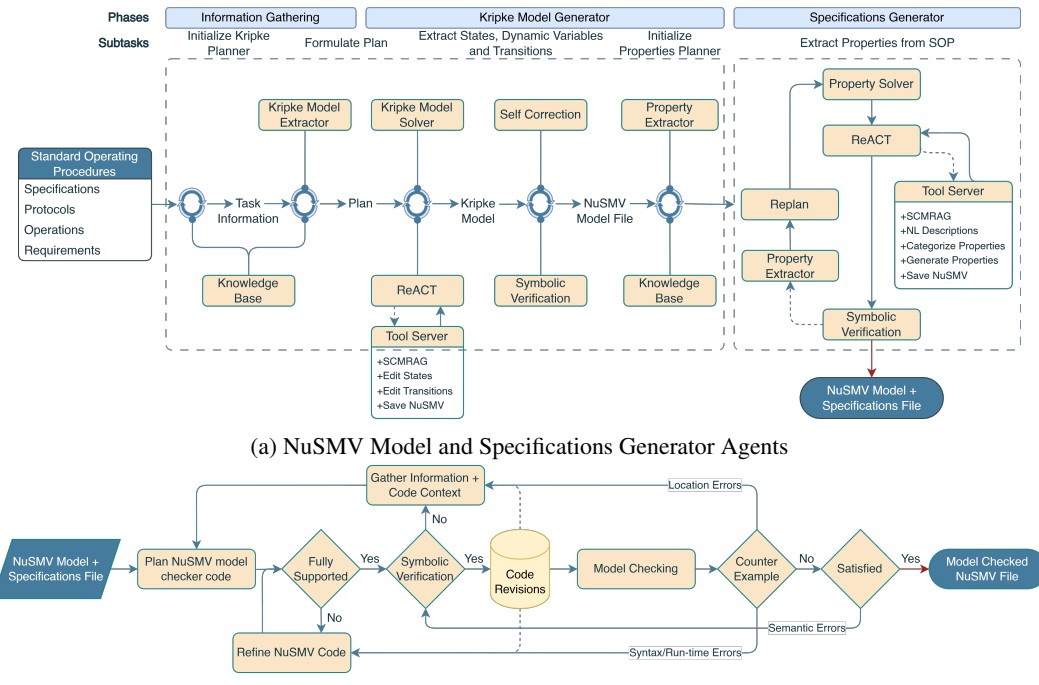

(a) NuSMV Model and Specifications Generator Agents

(b) NuSMV model-checker Coding and Debugging Agent

Figure 1: Overview of the SPECMAS framework, consisting of two pipelines: (I) **Model and Specification Synthesis**, where task information is extracted from the SOP to generate a Kripke structure and formal properties; and (II) **Model Checker Coding and Debugging**, which verifies the synthesized NuSMV model using temporal logic and iteratively refines it based on counterexamples or syntax errors until all properties are satisfied.

path (or tree) annotated with the falsified sub-formulas at the point of failure. So, if the verification fails, counter-examples are used by the proposed debugging agent to recursively revise the model until the relation $\mathcal{M} \models \varphi$ holds for all specified properties, thus ensuring correctness by construction.

## 3.1 Architecture Overview

As shown in Fig. 1, the proposed architecture is organised around two tightly-coupled *agentic* pipelines. The first, **Kripke-Model and Specification Synthesis**, processes the raw Standard Operating Procedure (SOP) corpus, extracts states, variables and transitions, and packages the resulting Kripke structure together with automatically derived CTL/LTL properties into two artefacts: a `.smv` model file and a companion specification file. This pipeline unfolds over three successive phases—*Information Gathering*, *Kripke Model Generation*, and *Specification Generation*. Throughout these phases every agent can query a shared knowledge base for retrieval-augmented prompts and invoke a reasoning LLM that performs on-the-fly symbolic checks, validating both the global plan and the intermediate outputs of each task.

The second pipeline, **Model-Checking and Debugging**, takes these intermediate artefacts as input, executes the NuSMV engine, and inspects any syntax errors or counter-example traces that NuSMV returns. Guided by this feedback, the pipeline iteratively adjusts both the state-transition model and the property set until the model checker confirms that every specification is satisfied. Its output is a *final, machine-verified NuSMV file*, thereby closing the loop from informal SOP text to formally validated code. During each refinement cycle we expose the complete code-revision history to the reasoning LLM, enabling it to explore alternative bounds, variable domains and transition guards when repairing syntax or localisation faults. Once these structural issues are resolved, the verifier performs a semantic pass; if any SOP requirement remains unmet, the agent replans and the debugging loop repeats until full compliance is achieved.

Table 2: LLM-Judge evaluation results across five benchmark systems. Each cell shows a metric or compliance score, with models grouped by task. Green highlights indicate the top three performers; red marks the bottom two. ∗ represents that the `.smv` artifact failed to execute after debugging.

| Model | Structural Alignment | Property Fidelity | Semantic Fidelity | Code Bonus | Code Compliance | Execution Compliance | Final Compliance Score |
|---|---|---|---|---|---|---|---|
| **BRP Protocol** | | | | | | | |
| SpecMAS | 5.66 | 6.84 | 5.50 | 0.38 | $0.55 \pm 0.02$ | $0.9 \pm 0.14$ | **0.76** $\pm 0.08$ |
| Claude [25] | 6.66 | 8.16 | 6.84 | 0.1 | $0.60 \pm 0.08$ | $0.38 \pm 0.04$ | $0.46 \pm 0.05$ |
| DeepSeek [26] | 5.00 | 5.16 | 5.34 | 0.2 | $0.45 \pm 0.01$ | $0.5 \pm 0.14$ | $0.48 \pm 0.08$ |
| Gemini [27] | 7.34 | 8.84 | 8.17 | 0.52 | $0.75 \pm 0.04$ | $0.0^* \pm 0$ | $0.30 \pm 0.01$ |
| Qwen [24] | 6.67 | 8.00 | 7.17 | -0.14 | $0.55 \pm 0.04$ | $0.16 \pm 0.04$ | $0.32 \pm 0.04$ |
| **Shuttle Autopilot Guidance System** | | | | | | | |
| SpecMAS | 4.84 | 5.17 | 5.66 | 0.19 | $0.45 \pm 0.1$ | $0.97 \pm 0.05$ | **0.76** $\pm 0.01$ |
| Claude | 5.66 | 6.50 | 6.00 | 0.25 | $0.53 \pm 0.08$ | $0.34 \pm 0.12$ | $0.42 \pm 0.1$ |
| DeepSeek | 4.33 | 4.33 | 5.00 | 0.03 | $0.37 \pm 0.07$ | $0.45 \pm 0.17$ | $0.42 \pm 0.07$ |
| Gemini | 4.83 | 5.66 | 5.00 | -0.23 | $0.36 \pm 0.04$ | $0.6 \pm 0$ | $0.51 \pm 0.01$ |
| Qwen | 5.16 | 5.16 | 5.33 | 0.20 | $0.45 \pm 0.04$ | $0.04 \pm 0.05$ | $0.20 \pm 0.04$ |
| **Mutual Exclusion** | | | | | | | |
| SpecMAS | 8.5 | 7.84 | 7.00 | 0.19 | $0.66 \pm 0.09$ | $0.85 \pm 0.07$ | $0.77 \pm 0.08$ |
| Claude | 9.66 | 9.5 | 9.34 | 0.02 | $0.76 \pm 0.13$ | $0.95 \pm 0.07$ | **0.88** $\pm 0.09$ |
| DeepSeek | 8.5 | 9.16 | 7.16 | 0.03 | $0.66 \pm 0.13$ | $1.0 \pm 0$ | $0.86 \pm 0.05$ |
| Gemini | 8 | 9.84 | 6.5 | 0.38 | $0.73 \pm 0.08$ | $0.85 \pm 0.07$ | $0.80 \pm 0.08$ |
| Qwen | 9.66 | 10.0 | 9.5 | 0.06 | $0.79 \pm 0.06$ | $0.1 \pm 0$ | $0.38 \pm 0.02$ |
| **Robot Controller** | | | | | | | |
| SpecMAS | 4.0 | 4.50 | 3.50 | 0.28 | $0.38 \pm 0.06$ | $0.90 \pm 0.07$ | **0.69** $\pm 0.07$ |
| Claude | 6.83 | 8.34 | 8.0 | 0.73 | $0.76 \pm 0.04$ | $0.0 \pm 0$ | $0.30 \pm 0.02$ |
| DeepSeek | 5.16 | 6.16 | 5.50 | -0.06 | $0.44 \pm 0.04$ | $0.64 \pm 0.06$ | $0.56 \pm 0.02$ |
| Gemini | 8.84 | 9.34 | 8.34 | 0.22 | $0.75 \pm 0.03$ | $0.0^* \pm 0$ | $0.30 \pm 0.01$ |
| Qwen | 6.66 | 7.17 | 5.66 | 0.34 | $0.58 \pm 0.05$ | $0.0^* \pm 0$ | $0.24 \pm 0.02$ |
| **GigaMax Cache Coherence Protocol** | | | | | | | |
| SpecMAS | 5.0 | 5.67 | 4.67 | 0.46 | $0.50 \pm 0.04$ | $0.84 \pm 0.06$ | **0.70** $\pm 0.05$ |
| Claude | 6.5 | 8.84 | 6.84 | 0.27 | $0.65 \pm 0.01$ | $0.06 \pm 0.04$ | $0.29 \pm 0.01$ |
| DeepSeek | 6.16 | 7.16 | 6.50 | 0.14 | $0.56 \pm 0.01$ | $0.66 \pm 0.05$ | $0.62 \pm 0.02$ |
| Gemini | 8.5 | 9.34 | 7.66 | 0.40 | $0.76 \pm 0.01$ | $0.0^* \pm 0$ | $0.30 \pm 0.01$ |
| Qwen | 6.5 | 7.16 | 6.84 | -0.08 | $0.53 \pm 0.04$ | $0.61 \pm 0.01$ | $0.57 \pm 0.01$ |

## 4 Experimental Setup

We report results on **MiniSpecBench**, a curated suite of 10 Standard-Operating-Procedure (SOP) documents paired with the reference expert NuSMV model files with CTL/LTL specifiactions. All expert written `.smv` files were taken from the canonical NuSMV example suite distributed with version 2.5.1 and later, covering a broad spectrum of industrial design motifs. Because the original distribution provides code but no documentation, we automatically synthesised a natural-language SOP for every NuSMV model using a customised variant of SYNTHIA [23], supplying the ground-truth `.smv` files as the sole input. SpecMAS queries the SOP documents and the available knowledge base (consisting of documents on formal verification; more information in supplementary material) during agent execution through a retrieval-augmented generation framework.

Results are averaged over two independent LLM-as-a-Judge evaluation runs. Full results on the MiniSpecBench dataset will be included in the supplementary material. The SpecMAS framework uses 8-bit-quantised Qwen3-32B model [24] as its backbone reasoning LLM. All experiments are conducted on a workstation equipped with an AMD Ryzen Threadripper PRO 7955WX, 256GB RAM and two NVIDIA RTX A6000 GPUs.

## 5 Evaluation

We evaluate SpecMAS against baseline LLM models using an LLM-as-a-Judge framework [28] for evaluation. Each generated `.smv` is assessed across three Code Quality metrics- **Structural Alignment**, **Property Fidelity**, **Semantic Fidelity**, along with a Bonus metric **Code Bonus**, and **Compliance Scores**.

Structural alignment captures how faithfully the agent reproduces the topology of the expert model and consists of the following components: (i) *role coverage* of critical variables, (ii) accuracy of *transition logic*, and (iii) consistency of the chosen *module/define* decomposition.

Property fidelity measures the logical equivalence and syntactic similarity between the generated properties and the expert-authored reference properties. It quantifies the quality of the generated CTL / LTL suite and comprises of: (i) *coverage* of the SOP- or expert-specified requirements, (ii) *logical equivalence* to expert formulas, and (iii) syntactic *operator correctness*. Semantic Fidelity evaluates behavioural soundness: (i) *behaviour match* under the execution semantics, (ii) treatment of *edge cases* such as recovery modes or fairness constraints, and (iii) clarity of identifier naming. Property and Semantic fidelity metrics are essential as generated properties can be logically sound and semantically well-formed, yet still lead to counterexample traces during model checking due to mismatches in assumptions, missing state conditions, or incorrect variable scoping.

Compliance Scores are mixtures of penalty and reward values that are used to measure the Model's adherence to the input SOP. (i) *Code Compliance* measures the model's overall coverage of input SOP requirement to NuSMV syntax and module definitions, (ii) *Execution Compliance* denotes the Model's functional coverage of input SOP in terms of property specifications and state variables and the ability to produce executable code without the presence of counter-example traces. The Bonus Metric evaluates a model's ability to generate a NuSMV model that surpasses expert-written files in quality by exploring additional edge cases, tightening specification bounds, and improving structural modularity beyond the reference implementation. Code Quality metrics are reported on a scale of 10, the Bonus and Compliance scores are in range $[-0.5, 1]$ and $[0, 1]$ respectively.

Table 2 presents a detailed evaluation of five systems from Mini SpecBench- Mutual Exclusion for two processes, BRP Message Protocol, Shuttle Autopilot Guidance System, Real-time robot controller, GigaMax Cache Coherence Algorithm. Each system was assessed across a total of 17 metrics, which were aggregated into higher-level categories for analysis as described above (implementation details are provided in supplementary material). The Final Compliance Scores are then calculated as the weighted average of the Code Compliance ($40\%$) and Execution Compliance ($60\%$) scores.

The evaluation results highlight significant variations in performance across all models and metrics, reflecting the diverse strengths and limitations of each system in the task of generating executable and semantically aligned NuSMV code from SOP documents. Among all models tested, SpecMAS, Claude, and DeepSeek consistently perform better overall, particularly in terms of structural and semantic fidelity. Claude (3.7-Sonnet) performs competitively, particularly in structural alignment and property fidelity, closely trailing SpecMAS on several benchmarks. Both Claude and SpecMAS achieve high Semantic fidelity score, revealing the better instruction following and CoT-based exploration capabilities of these methods. However, Claude's performance drops significantly on execution-sensitive tasks, as seen in its failure to execute the Robot Controller model.

DeepSeek (R1-Distill-Llama-70B) maintains balanced, moderate performance across metrics, ranking third overall, but occasionally struggles with semantic precision in complex safety-critical systems. Gemini (Flash-2.5), despite scoring well on some logical and semantic aspects, suffers from poor execution reliability and fails to generate executable models in three out of five tasks. This reflects structural inconsistencies and weaker domain alignment. Surprisingly, Qwen (3.0-Chat-235B), despite having a much larger parameter count than SpecMAS's Qwen3-32B 8bit model, performs poorly across most tasks particularly in execution. This suggests that SpecMAS's advantage does not come from model size or raw reasoning power alone. Instead, its strength lies in the integration of model and specification synthesis with an iterative code debugging pipeline.

SpecMAS demonstrates the most stable and robust performance across the five benchmark systems, achieving the highest Final Compliance Score for four out of five SOPs evaluated in Table 2. It combines strong formal reasoning, structured synthesis, and execution reliability in a single unified framework. It further maintains a strong balance across all evaluation axes: structural alignment, property fidelity, semantic fidelity, and execution correctness. This indicates its superior ability to generalise system behaviours from SOPs into valid formal models. Notably, SpecMAS excels in Execution Compliance, outperforming all other models on this critical dimension.

Table 3: Comparison of lowest Debug-Step models vs. highest Final Compliance score models.

| SOP | Min Debug-Steps | | Max Final-Compliance Score | | △ Debug Steps | △ Score |
|---|---|---|---|---|---|---|
| BRP Protocol | DeepSeek | 1 | SpecMAS | 0.76 | +5 | +0.28 |
| Shuttle Autopilot | DeepSeek | 0 | SpecMAS | 0.76 | +5 | +0.34 |
| Mutual Exclusion | Claude | 0 | SpecMAS | 0.77 | +6 | -0.11 |
| Robot Controller | Claude | 1 | SpecMAS | 0.69 | +4 | +0.39 |
| GigaMax Cache | DeepSeek | 2 | SpecMAS | 0.7 | +3 | +0.08 |

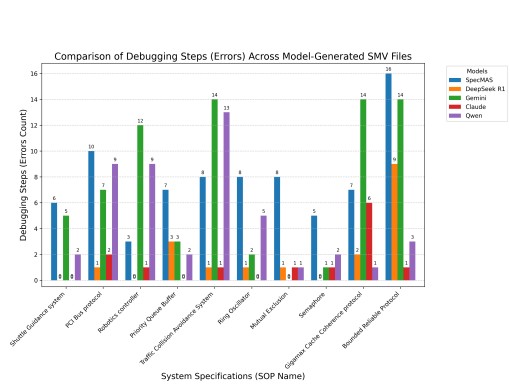

(a) Comparison of debugging/error correction steps required to execute generated `.smv` files.

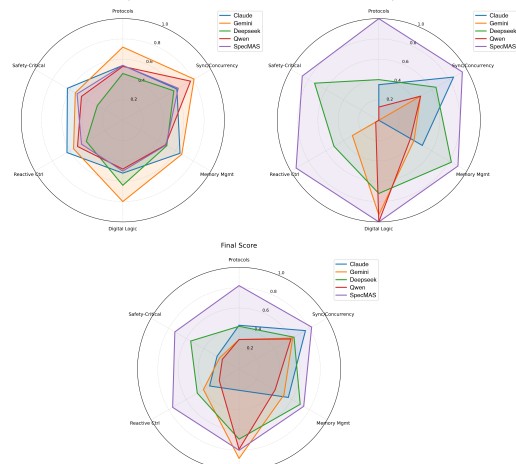

(b) Comparison of compliance scores for `.smv` files of different SOP system domains in MiniSpecBench.

Figure 2: Evaluation of generation performance across debugging effort and compliance quality.

## 5.1 Qualitative Analysis of Code Generation and Execution Compliance

For all models except SpecMAS, we measure the number of manual debug steps-defined as the minimal edits required to produce a syntax-error-free file. For SpecMAS, we instead track the number of iterations performed by our Debugger Agent, which first resolves syntax errors and then iteratively refines counterexamples until a model-checked artifact is produced. Figure 2a compares the number of manual debug steps and agent-led repair steps across ten benchmark SOPs. Intuitively, fewer debug steps suggest that an LLM's initial draft is closer to a "compile-and-verify" SMV artifact.

However, as shown in Tables 2 and 3, once all files are repaired, SpecMAS consistently outperforms all baselines on key metrics such as Structural Alignment, Property Fidelity, and Compliance Scores. This highlights a critical insight: low debug step counts do not necessarily translate into high-quality or high-compliance `.smv` files. A plausible explanation is that baseline models may produce files that are syntactically valid with minimal edits (or none at all), but structurally incomplete or semantically shallow. For instance, DeepSeek requires zero syntax fixes but achieves very low Final Compliance Scores. This suggests that, while its code is "runnable", it often omits critical state variables, transition logic, or property specifications. In contrast, SpecMAS generates drafts that are more detailed and semantically rich-capturing edge cases, nuanced transitions, and full property coverage-which, while increasing the likelihood of syntax errors. But this results in much higher semantic fidelity once agentic debugging is applied as shown in Figure 2b.

The tendency of baseline LLMs to produce error-prone `.smv` code is likely linked to NuSMV being a low-resource, domain-specific language, where token-level exposure during pretraining is minimal. This leads to higher hallucination rates, a consistent pattern observed especially in Gemini and Qwen. These issues reinforce the value of providing LLMs with tools and domain-specific context during generation. Our evaluation strongly supports this hypothesis: tool-augmented, context-aware models (like SpecMAS) produce higher-quality artifacts, even if they initially require more debugging effort.

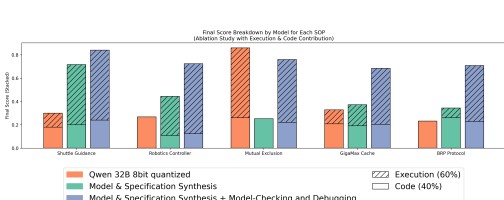

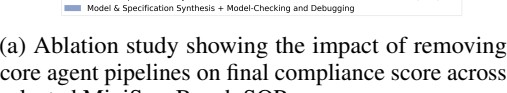

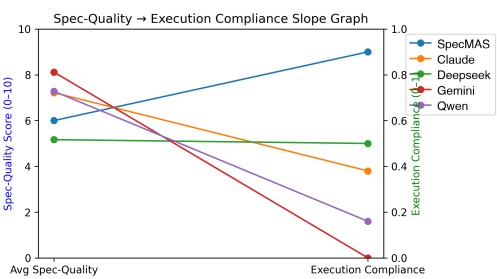

(a) Ablation study showing the impact of removing core agent pipelines on final compliance score across selected MiniSpecBench SOPs.

(b) Case study comparing average specification quality (left axis) and execution compliance (right axis) across models for the BRP Protocol SOP.

Figure 3: (a) Contribution of individual agents in SpecMAS to overall model-checking performance. (b) Analysis of the trade-off between Specification quality (SPEC) and Execution Compliance.

## 5.2 Ablation Studies for Evaluation of Agent Contributions

The ablation study results shown in Figure 3a evaluate the incremental contribution of key components within the SpecMAS pipeline, specifically the impact of: (i) The baseline Qwen3 32B 8-bit LLM without planning or verification model, (ii) Model and specifications synthesis alone (iii) the full pipeline including symbolic model checking and debugging—on the Final Compliance score across five benchmark SOPs from Table 2. Each axes is decomposed into two stacked segments representing the weighted contributions of Code Quality and Execution Compliance to the final score.

The ablation results confirm that the key performance gains of SpecMAS are not solely from LLM-based synthesis, but from its integrated verification and debugging loop. While high-quality initial generation is important, reliable execution, particularly in formal verification tasks, requires post-hoc error correction and planning. This is a feature that the full SpecMAS pipeline provides. Synthesis-only pipeline improves code compliance over baseline Qwen model and often fails to produce executable models. For instance, in the Robotics Controller and GigaMax Cache benchmarks, execution performance without debugging counterexample traces remains low.

We observe a drop in performance for the Mutual Exclusion SOP when using only the model and specification synthesis pipeline. The intermediate artefact generated at this stage was not executable due to syntax errors and a missed explicit state transition condition. Additionally, the property exploration plan introduced a novel property related to turn alternation which was not explicitly required by the SOP. Once the agentic debugging loop was activated, it successfully identified and corrected both the structural and property-level issues.

## 5.3 Case Study: Evaluating the Impact of Proposed Agents

We observe that although SpecMAS occasionally scores lower than baseline LLMs like Claude and Gemini on structural alignment, property fidelity, and semantic fidelity, it still achieves the highest Final Compliance scores. This raises a question about how lower Code Compliance artifacts can outperform more aligned model files in Execution Compliance. We attribute this to two main factors: (1) the execution-centric design of SpecMAS and (2) the brittleness of one-shot generation in baseline models. To illustrate this, we examine the BRP protocol SOP from MiniSpecBench.

Figure 3b highlights this paradox. It plots Execution Compliance against Average Specification (SPEC) quality for BRP Protocol SOP, calculated as the mean of structural alignment, property fidelity, and semantic fidelity. Although Qwen achieves the highest mean SPEC quality score, it fails almost completely in Execution Compliance similar trend can is observed for Gemini as well. In contrast, SpecMAS and Claude have more modest average SPEC quality scores, but maintain a high Execution Compliance.

This discrepancy reveals a critical flaw in one-shot generation approaches. While their outputs may appear structurally or logically correct, they lack post-generation verification, counterexample-guided refinement, and SOP-grounded planning. As a result, even minor syntax or semantic issues that are undetectable by surface-level scoring can lead to failure during NuSMV execution. In contrast,

SpecMAS deliberately trades alignment and fidelity scores to incorporate multi-agent planning and automated debugging pipelines that corrects errors before finalizing the model file. This result in lower Code Compliance score, but a substantially higher Execution Compliance—an outcome that is far more valuable for formal verification tasks.

# 6 Conclusion

In this work, we introduced SpecMAS, a multi-agent framework for generating formally verified systems from natural language SOPs using model checking. Through evaluations on the MiniSpecBench suite, SpecMAS consistently outperformed state-of-the-art LLMs across key metrics. Our results highlight the value of an execution-centric, multi-agent approach over one-shot code generation. By combining planning, retrieval, and automated debugging, SpecMAS improves the reliability of generated NuSMV models. This work marks a step forward in applying agentic LLMs to system synthesis and verification, paving the way for more robust LLM based formal verification agents.

# 7 Limitations and Future Directions

While SpecMAS demonstrates strong potential for formal self-verification of agentic systems, some limitations remain that highlight valuable opportunities for future research and refinement.

First, handling underspecified or inconsistent SOPs is inherently challenging due to the variability and ambiguity of real-world documentation. SpecMAS already mitigates this through reasoning-capable LLMs, hierarchical planning, and SCMRAG [29] based retrieval to infer missing information and align transitions with the logical intent of the SOP. However, extreme ambiguity may still require human-in-the-loop assistance. Integrating guided expert feedback into the verification loop is an extension that would enhance semantic fidelity and interpretability without compromising automation.

Second, while the system's property extraction and symbolic relevance checks substantially reduce vacuous or trivial properties, there remains limited risk that certain verified properties may not fully capture the intended operational semantics. The built-in entailment classification and counterexample-based debugging already provide interpretability and grounding, but future versions can incorporate richer semantic reasoning and coverage metrics to strengthen this safeguard further.

Third, SpecMAS currently relies on NuSMV for symbolic model checking, which restricts full generalization to hybrid or probabilistic systems. Nonetheless, our results on the traffic collision avoidance and shuttle autopilot controller tasks show that the framework can model continuous behaviors through discrete abstractions. Future expansions will integrate complementary verification backends such as SPIN, extending the approach to hybrid and stochastic systems.

Finally, scalability is an ongoing consideration. Symbolic model checking can encounter state-space explosion in high-complexity systems, occasionally increasing reasoning time and debugging iterations. Current results show strong convergence, with an average of only 18 additional steps on the most complex tasks. Future work will explore component-based and modular verification, allowing larger systems to be decomposed and verified more efficiently. Also, the current implementation of SpecMAS focuses on generated SOPs rather than human-authored documents. This choice was made because existing NuSMV model files lacked corresponding natural-language specifications, making aligned SOP-model pairs unavailable. SYNTHIA was designed to generate semantically faithful yet syntactically diverse SOPs from human-authored code artifacts. We plan to incorporate human-written SOPs in future work to assess robustness under natural linguistic variability.

In summary, these limitations represent growth avenues rather than shortcomings, each pointing toward richer semantic understanding, broader generalizability, and improved scalability. Addressing them will further strengthen SpecMAS as a reliable, interpretable, and extensible framework for formal self-verification of LLM-generated system models.

# 8 Acknowledgments and Disclosure of Funding

This research was funded through the MITACS Accelerate Program (Grant IT40058) in partnership with the International Center for Applied Systems Science for Sustainable Development(ICASSSD), Cambridge, Ontario.

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
