# OpenReview forum: "SpecMAS: A Multi-Agent System for Self-Verifying System Generation via Formal Model Checking"
_NeurIPS.cc/2025/Conference — NeurIPS 2025 poster_

### Official Review · Reviewer_hC91 · 2025-06-22

**Clarity:** 4
**Significance:** 4
**Originality:** 4
**Rating:** 5
**Confidence:** 4

**Summary:**

The paper proposes a general methodology for generating exacutable systems from NL specification and uses symbolic model checking, and the NuSMV tool in particular, to validate the systems generated

**Questions:**

Why are you bringing multiagents into your paper? I understand that this is your general long-term reference application scenario. But, it seems to me, that all you have done is largely independent of this. All the work you have done applies to single FSM, possibly composed from simpler FSMs

Your citation of NuSMV is wrong. If you type NuSMV to scholar you will find appropriate citations starting from the very early paper(s)

**Ethical Concerns:**

["NO or VERY MINOR ethics concerns only"]

**Final Justification:**

In my opinion, this paper should definitely be accepted. It is a first serious attempt to deal with the problem of how to understand an informal input correctly. It draws on state-of-the-art formal methods techniques, and it applies them in this context. It is a first step in a promising direction which can bring the community a new way of dealing with the general problem of hallucinations and the like.

The process of going from informal to formal specifications is extensively studied in formal methods. And this is another good reason why this paper should be accepted, as it opens the way to more cross-fertilisation. At the same time, the domain of application is new and much more complex (in terms of ambiguity of the input) with respect to the mainstream work on formal methods. Here, in fact, the input is kept as simple as possible in order to minimise mistakes.

All in all, this work paves the way for possible cross-fertilisation between two communities which are quite far apart but have a lot to say to each other.

**Limitations:**

yes

**Quality:**

3

**Strengths And Weaknesses:**

strengths
- The approach is novel. It is not an incremental evolution of the SoA
- the work is a concete tentative to generate systems which are highly reliable, by building on the extensive work done in the formal methods community

Weaknesses
- no major weaknesses

---

> ### Author Rebuttal · Authors · 2025-07-31
>
> # Rebuttal – Reviewer hC91
>
> We appreciate the reviewer’s observation and the opportunity to clarify the role of the multi-agent design in our framework. While it is true that the final output of **SpecMAS** is a formally verified finite-state model (FSM), the multi-agent architecture is central to how the finite-state model is constructed, verified, debugged and refined from natural language input.
>
> ## **SpecMAS** consists of functionally distinct agents, each responsible for a critical stage:
>
> 1. Parsing and reasoning over SOPs
> 2. Extracting states and transitions
> 3. Inferring explicit and implicit temporal properties
> 4. Verifying and debugging the model using NuSMV counterexamples
>
> These agents coordinate via shared memory and planning, and their outputs are verified and refined in a closed-loop manner. This agent-based decomposition allows **modularity**, **specialization**, and **recursive self-correction**, which are difficult to achieve in a single-agent or monolithic system.
>
> Moreover, the proposed design enables **scalability** and **generalisability**. For instance, in future versions, **SpecMAS** can be specialized to handle hybrid systems, different model checkers, or real-time constraints without modifying the entire architecture. We believe this structured, agent-oriented approach is critical for bridging unstructured natural language specifications with the requirements of formal verification.
>
> We will revise the citation of NuSMV and include appropriate citations for the relevant earlier papers as well.

---

> ### Comment · Reviewer_hC91 · 2025-08-01
>
> The answer provided satisfies my request. I confirm my acceptance evaluation.

---

> > ### Comment · Reviewer_hC91 · 2025-08-02
> >
> > As it was correctly noted, this paper concentrates on the formalisation step, rather than on the reasoning on the formalised problem. This is actually to me what makes this paper most interesting and novel.

---

> > > ### Author Response · Authors · 2025-08-07
> > >
> > > Thank you for your constructive feedback. We appreciate your time and look forward to this work making a meaningful contribution.

---

### Official Review · Reviewer_5uhY · 2025-07-02

**Clarity:** 3
**Significance:** 3
**Originality:** 3
**Rating:** 4
**Confidence:** 4

**Summary:**

This paper describes the use of LLMs to synthesize finite state models from informal natural language descriptions. These models are amenable to automatic verification using model checking. The approach uses the NuSMV tool for verification with respect to temporal properties, which are also extracted from the documentation. The models are iteratively refined by the LLMs with the help of counterexamples obtained from NuSMV. An evaluation is presented, using several LLMs using 10 mini benchmarks created by the authors.

**Questions:**

Please provide a detailed description of: prompts, benchmark sizes, properties, and metrics. Adding these details would greatly strengthen the paper.

**Ethical Concerns:**

["NO or VERY MINOR ethics concerns only"]

**Limitations:**

yes

**Quality:**

3

**Strengths And Weaknesses:**

Pros:

*    This work is in the same line as verifiable code generation with LLMs but it targets higher-level models that are amenable to formal verification with NuSMV, This seems to be a new, promising target application for LLM generation.

* The approach is well designed: a reasoning agent extracts both explicit and implicit properties from the system description, verification is performed via model checking and another debugging agent analyzes counterexamples and iteratively corrects the model until all properties are satisfied.

*    Results show that the use of model checking produces better results than "self-verification" with LLMs.

Cons:

*    The line of work is very promising but the paper seems premature; it is written at a very high level with many details missing. It is not clear what prompts were used and also if any few shot examples were used to "help" with he process. Was any fine tuning necessary?

*    It is not clear what the size is for each generated model.

*    The paper is also missing a description of the properties that were verified. Were there any properties that would hold vacuously?

*    It is unclear what metrics were used for Table 2. They are described informally without a precise definition.

---

> ### Author Rebuttal · Authors · 2025-07-31
>
> # Rebuttal – Reviewer 5uhY
>
> We thank the reviewer for recognising the promising nature of our work and your constructive suggestions for further improvement. We have addressed your questions below.
>
> ## Prompts and Few-Shot Examples
>
> - The full prompt templates used by each agent are provided in **Section 5** of the supplementary material, and additional prompts are covered in the `Code` folder of the zip file with the suffix `*_templates.py`. We also provide the entire **MiniSpecBench** dataset in the supplementary zip file. We acknowledge that including specifics on property definitions and evaluation metrics would further strengthen the paper. To address this, we have included the relevant source code in the supplementary material and we will expand it with implementation details in the supplementary section of the final version.
>
> - Our framework does not use few-shot examples in the traditional sense. Instead, it leverages **SCMRAG** to retrieve contextual knowledge from a knowledge base, which includes NuSMV documentation and model checking principles (see **Section 4** of the supplementary material). For instance, when extracting states and dynamic variables, the Model Generator Agent may request background knowledge on Kripke structures, and formal methods for representing system states in the information gathering phase. This enables informed and accurate model synthesis.
>
> ## Model Configuration
>
> - We confirm that no model fine-tuning was performed. Our framework utilizes the off-the-shelf **Qwen3 32B** model in reasoning mode, quantized to 8 bits.
>
> ## Benchmark Sizes and Metrics
>
> - The average file size for our SOP documents is **32.7 KB**. As for generated NuSMV models, the minimum file size is **3 KB** and the maximum is **8 KB** (BRP protocol), with an average of ~14 **SPEC** properties and a maximum of 20 (priority-queue model), representing realistic complexity for many safety-critical systems.
>
> - We verify standard temporal logic properties generated by the **Specification Generator Agent**. This agent systematically analyzes the input SOP to identify both explicitly stated properties and implicitly required properties related to **safety**, **liveness**, **fairness**, **reachability**, and **temporal ordering** (see `extraction_templates.py` in the supplementary material for prompt details). The resulting candidate properties are then passed to the **Debugging Agent**, which iteratively verifies them using **NuSMV** until all counterexample traces are resolved.
>
>   While some properties may hold trivially if directly requested in the SOP, the agent does not generate vacuous properties. Our **Property Extractor Agent** is explicitly designed to extract semantically meaningful properties that are grounded in the system’s operational logic. Furthermore, since our SOPs are derived from expert-written NuSMV models, the extracted properties retain both semantic relevance and verification significance, ensuring that the verification process remains aligned with the system’s intended behavior.
>
> - We have discussed the metrics in **Section 5** of the paper and provide the source code for evaluating all metrics in the supplementary material. However, we acknowledge the necessity of formal definitions for relevant metrics and will include them in the supplementary section of the final version.
>
> ---
>
> We appreciate your assessment that this work is promising and will incorporate these detailed specifications to strengthen the paper's technical contributions and reproducibility.

---

> > ### Comment · Reviewer_5uhY · 2025-08-01
> >
> > Thank you for the response.
> >
> > You say "the extracted properties retain both semantic relevance and verification significance, ensuring that the verification process remains aligned with the system’s intended behavior."
> >
> > While I understand this is desired I get that it is not guaranteed that the extracted properties have semantic relevance. For instance all of them can vacuously hold on the model leading to no counterexamples. As a result of such check, you would not know if the model is correct. In that case you would need to repair the properties rather than the model. How do you handle such cases?

---

> > > ### Author Response · Authors · 2025-08-01
> > >
> > > Thank you for the response. We agree that the absence of counterexamples alone does not guarantee that a property is semantically relevant to the intended behavior described in the SOP. To address this, SpecMAS incorporates an explicit symbolic relevance check as part of the debugging pipeline (see Figure 1b, node “Satisfied”). For each verified property, the Debugging Agent is prompted to assess its alignment with the SOP through a *semantic entailment classification*, using one of four labels: *["Fully supported", "Partially supported", "No support", "Contradictory"]*. This ensures that properties are not only syntactically valid and verifiable but also meaningfully grounded in the specification.
> > >
> > > This entailment reasoning is performed using instruction prompt detailed in `knowledge_search_templates.py` in the supplementary code. If a property is found to be partially supported, unsupported or contradictory, the agent can autonomously choose to repair the property using the generated reasoning as feedback or plan a new candidate that better aligns with the SOP. This process is reflected in the feedback loop of Figure 1b.
> > >
> > > We acknowledge that this does not eliminate all risks of vacuous satisfaction, but it introduces a practical and interpretable mechanism to detect and mitigate such cases, thereby improving the robustness of the verification process.

---

> > > > ### Comment · Reviewer_5uhY · 2025-08-04
> > > >
> > > > Thank you for your answer.
> > > > May I please ask: how do you make sure that the generated transition system and the specification refer to the same variables?

---

> > > > > ### Author Response · Authors · 2025-08-04
> > > > >
> > > > > Ensuring consistency between the variables used in the transition system and those referenced in the generated specifications is an important aspect of SpecMAS. It is addressed through a shared intermediate representation in built during the Kripke Model Generation phase that encodes the system's states (S), initial states (I), transition relations (R), and labeling function (L). Both the Model Generator Agent and the Specification Generator Agent operate over this unified representation, which contains structured metadata extracted from the SOP. This shared representation guarantees that the specification logic (e.g., in CTL/LTL formulas) is grounded in the same variable set as the model’s transition definitions.

---

> > > > > > ### Author Response · Authors · 2025-08-07
> > > > > >
> > > > > > Responding to reviewer's question:
> > > > > > Were there significant challenges in translating from natural language to CTL and LTL?
> > > > > >
> > > > > > Yes, translating from natural language to CTL and LTL posed several significant challenges, particularly due to the inherent ambiguity and underspecification present in many SOPs. Following are a few prominent ones we faced:
> > > > > >
> > > > > > 1. Temporal logic constructs such as safety ("something bad never happens") and liveness ("something good eventually happens") are often described implicitly in natural language without precise temporal cues. Capturing these abstract notions in formal logic required not only syntactic parsing but also semantic inference, which we addressed using a reasoning LLM guided by retrieval-augmented context from formal verification literature.
> > > > > >
> > > > > > 2. The mapping from SOP statements to correct quantifiers and temporal operators is non-trivial. It often depends on subtle aspects of system behavior that are not explicitly stated. To handle this, the Specification Generator Agent leverages a structured prompting mechanism that first identifies relevant behavioral patterns and then formulates candidate properties accordingly.
> > > > > >
> > > > > > 3. Maintaining variable consistency between the transition model and the logical specification required careful coordination through a shared intermediate representation. Without this alignment, though properties may be syntactically valid, they would fail during NuSMV model checking due to undefined or mismatched variables.
> > > > > >
> > > > > > 4. Even if a property is syntactically well-formed and passes verification (i.e., no counterexamples are found), it may still be semantically misaligned with the SOP. To detect such cases, we introduce an additional property verification step where the Debugging Agent performs entailment classification based on the SOP context.
> > > > > >
> > > > > > These challenges highlight why the proposed SpecMAS multi-agent framework is necessary to produce semantically meaningful and verifiable specifications from natural language descriptions.

---

### Official Review · Reviewer_vGPi · 2025-07-02

**Clarity:** 3
**Significance:** 3
**Originality:** 3
**Rating:** 5
**Confidence:** 4

**Summary:**

SpecMAS presents a multi-agent framework that translates natural language Standard Operating Procedures (SOPs) into formally verified NuSMV models. It features two tightly coupled pipelines: a synthesis agent that generates models and specifications, and a debugging agent that uses model checking feedback to iteratively refine the output until all properties are satisfied. Evaluation on the MiniSpecBench dataset shows that SpecMAS outperforms LLM baselines across several compliance and semantic fidelity metrics.

**Questions:**

How would SpecMAS handle inconsistent or underspecified SOPs in real-world documentation?

Can the approach generalize to hybrid systems (e.g., with continuous dynamics)? Could the authors release MiniSpecBench to foster benchmarking standardization?

Are there plans to extend the framework to support probabilistic or temporal stochastic model checkers? What about the state space explosion problem with nusmv models?
Can you quantify the performance cost of the multi-agent debugging loop in high-complexity specifications?

**Ethical Concerns:**

["NO or VERY MINOR ethics concerns only"]

**Final Justification:**

I am maintaining my score of 5 (Accept).

The paper presents a novel, well-executed approach for bridging natural language SOPs and formal verification through an agent-based, closed-loop architecture. It fills a significant gap between NLP and formal methods, and the rebuttal reinforces the contribution's robustness, generality, and future potential.

**Limitations:**

No major concerns. The paper explicitly addresses the synthetic SOP generation process, the limitations of NuSMV, and trade-offs in fidelity versus execution.

**Quality:**

3

**Strengths And Weaknesses:**

Strengths:
First work to propose an agent-based closed-loop pipeline for formal model generation from natural language using NuSMV. Strong architecture, comprehensive experiments, robust benchmarking across multiple domains.

Offers a scalable and generalizable framework that can bridge natural language and formal verification – a key need in system synthesis. The writing is dense but clear, with well-annotated diagrams and detailed evaluations.

Weakness:
Reliance on NuSMV may limit expressivity in modeling probabilistic or timed systems.

SOPs are synthetically generated for some benchmarks, raising concerns about real-world applicability. The debugging loop might struggle with ambiguous or highly underspecified SOPs – not thoroughly evaluated.

---

> ### Author Rebuttal · Authors · 2025-07-31
>
> # Rebuttal – Reviewer vGPi
>
> Thank you for your positive assessment and recognition of our framework’s contributions. We appreciate your insightful questions about extensions and limitations.
>
> ---
>
> ### Question: “How would SpecMAS handle inconsistent or underspecified SOPs in real-world documentation?”
>
> This is a critical point and one of the central challenges SpecMAS is designed to address. In its current form, SpecMAS leverages a reasoning-capable LLM, hierarchical planning, and SCMRAG-based context retrieval to resolve ambiguities present in the SOP. Importantly, the Specification and Model Generator agents are capable of extracting both explicitly stated and implicitly inferred properties and transitions, ensuring alignment with the underlying logical structure of the SOP, even when parts of it are underspecified. That said, we acknowledge that extreme ambiguity may still require external resolution, and in future work, we plan to explore incorporating a human-in-the-loop component to assist in such cases while maintaining interpretability in the model generation process.
>
> ---
>
> ### Question: “Can the approach generalize to hybrid systems (e.g., with continuous dynamics)?”
>
> We agree that our current reliance on NuSMV limits the framework to continuous and non-probabilistic systems. We chose NuSMV as it is a mature, industry-standard symbolic model checker, making it an ideal starting point. A natural direction for future work would be to extend the framework to support other verification engines, such as PRISM, TLA+, SPIN etc by creating new toolsets for these agents.
>
> SpecMAS does generalize to systems with continuous dynamics as illustrated by the (Traffic collision avoidance) and (Shuttle autopilot controller) examples in the Benchmark dataset. These systems model the continuous behaviour of traffic or aircraft guidance systems with dynamic variables.
>
> ---
>
> ### MiniSpecBench Release
>
> We plan to publicly release the proposed MiniSpecBench dataset, including the SOPs, expert models and evaluation scripts at a later date. These files are provided in the supplementary material of the submission.
>
> ---
>
> ### State Space Explosion
>
> We recognize that symbolic model checking can suffer from state space explosion, especially for large systems. For instance, in our experiments with a deadlock detection algorithm containing an extensive number of state variables, SpecMAS required significantly more reasoning time and a higher number of debugging iterations to resolve all counterexample traces. To address this, we plan to explore component-based verification techniques in future work. This would enable SpecMAS to decompose large systems into smaller, independently verifiable modules and improve scalability without compromising correctness.
>
> We conducted a comparative analysis of examples with the highest Final Compliance Scores relative to the number of debugging steps (see Table 3). However, we have not yet quantified the performance cost of the multi-agent debugging loop for high-complexity specifications, as our current benchmark does not explicitly distinguish SOPs of that category. On average, the debugging agent takes 18 steps more to debug highly complex specifications. We will consider including more detailed analysis of debugging agent in future work.

---

> > ### Comment · Reviewer_vGPi · 2025-08-01
> >
> > I thank the authors for their detailed and thoughtful rebuttal. I summarize my updated thoughts below:
> >
> > Underspecified SOPs
> >
> > The explanation of how SpecMAS leverages SCMRAG-based retrieval and LLM-driven inference to handle underspecified procedures is reassuring. I appreciate the authors’ acknowledgment that extreme ambiguity may still require human intervention and agree that a human-in-the-loop model could be a valuable extension.
> >
> > Generalization to Hybrid Systems
> >
> > The current NuSMV-based backend constrains expressivity, but the authors' discussion of extending support to other backends (e.g., PRISM, SPIN, TLA+) is promising. The use of hybrid domains like traffic and guidance systems in the current benchmarks is a helpful indicator of broader applicability, though formal hybrid system modeling remains future work.
> >
> > Dataset Release
> >
> > Good to see that MiniSpecBench will be released. This will benefit the community and support benchmarking in this emerging area.
> >
> > Scalability and Debugging Overhead
> >
> > The discussion of state space explosion and debugging iteration counts is informative. While a more detailed quantitative analysis of runtime overhead in high-complexity cases would strengthen the work, I appreciate the authors’ transparency and direction toward component-based verification.

---

### Official Review · Reviewer_YKJx · 2025-07-03

**Clarity:** 1
**Significance:** 2
**Originality:** 3
**Rating:** 3
**Confidence:** 3

**Summary:**

This paper presents a framework for synthesizing and verifying programs from natural language descriptions (Standard Operating Procedure). The framework represents the generated system as a finite-state transition system along with CTL/LTL specifications. The framework integrates LLMs and verification tools. In particular, LLMs are used to parse SOP, generate specifications, and make corrections. The paper demonstrates that SpecMAS outperforms program synthesis approaches that directly invoke a language model across a number of benchmarks over several metrics.

**Questions:**

- How is the number of minimal edits computed? Is it provably the minimal number of edits, or is it the empirical number of edits made by some human?
- Is SpecMAS fully automated?
- I imagine that the LLM was trained less on NuSMV text than other more common languages (e.g., C, Python). Have the authors encountered cases where the LLM has trouble generating syntactically correct models?

**Ethical Concerns:**

["NO or VERY MINOR ethics concerns only"]

**Limitations:**

The checklist suggests that there will be a discussion of limitations in the supplementary materials, but I did not find such discussion there.

**Quality:**

2

**Strengths And Weaknesses:**

Strengths
- This paper presents a framework that combines the strengths of LLMs and formal verification tools to generate programs that provably satisfy the extracted specifications. The framework is novel as far as I can tell.
- Leveraging existing framework like NuSMV seems like a good design choice.

Weaknesses:
- Ultimately, any approach that goes from natural language to formal language lacks formal guarantees on the correctness of the generated program. This work is no exception. This makes the use of formal verification techniques (which are expensive processes) less well-motivated.
- In order to improve the trust in the generated program, it seems the correctness of the synthesized specifications is important, otherwise the verification would not be meaningful. The paper lacks a rigorous analysis on the correctness of the generated specifications. What comes close to it is the measurement of Property Fidelity. However, SpecMAS is not competitive on this measure compared to baseline methods.
- The main methodology section is only a little over 2 pages, and stays at a very high-level. This makes it really difficult to fully grasp the workflow. I understand there is a page limit, but in its current form, the methodology reads quite thin and abstract. Walking through those steps with a working example would greatly improve the readability of the paper.
- The benchmarks are generated from existing NuSMV examples via SYNTHIA (not real human-written SOPs). This may lead to unrealistically clean natural language inputs. It would be interesting to test on existing natural language benchmarks or SOPs written by humans.

---

> ### Author Rebuttal · Authors · 2025-07-31
>
> # Rebuttal – Reviewer YKJx
>
> We thank the reviewer for their thorough reading and insightful suggestions. Their feedback already strengthens SpecMAS. Below are each point addressed in turn, with references to the sections in the paper or supplementary section wherever possible.
>
> ---
>
> ## 1) Formal Guarantees and Specification Correctness
>
> The core contribution of SpecMAS is to address the issue of verifying LLM generated system outputs through a closed-loop verification process. While it is true that no natural language-to-code system can offer a priori guarantees on correctness,
> SpecMAS ensures correctness of extracted properties through iterative model checking.
>
> As outlined in Section 3.1, SpecMAS avoids the pitfalls of one-shot synthesis by generating NuSMV specification files that are aligned with both explicitly stated and implicitly inferred logical constraints. Crucially, when a specification fails during model checking, SpecMAS automatically interprets the counterexample traces returned by NuSMV and uses them to refine the system model and specifications without requiring human intervention. This loop continues until all counterexamples are eliminated and the model satisfies all properties. Thus, even when the initial specifications are noisy or incomplete, the Model Generator, Coding, and Debugging Agents work collaboratively to correct structural and logical errors in an automated, iterative manner.
>
> The final output received from the SpecMAS system is machine-verified. This makes the use of formal verification both justified and essential. It elevates the reliability of a stochastic generative process into a provably correct artifact. This is also reflected in our evaluation where SpecMAS consistently produces executable and verified NuSMV model files. In contrast, larger LLMs often fail to generate functionally correct artifacts even after multiple debugging attempts.
>
> ---
>
> ## 2) Trust in Generated Specifications
>
> We agree that the correctness of the synthesized specifications is central to meaningful verification. Property Fidelity (defined in Section 5) is our primary metric for evaluating this aspect. While SpecMAS may not always achieve the highest Property Fidelity score across all systems, SpecMAS achieves the best Final Compliance Scores on 4 out of 5 benchmarks. This metric includes Execution Compliance which is a more rigorous indicator of whether the generated specifications are actually valid and verified, and not just syntactically plausible.
>
> Our case study (Section 5.4, Figure 3b) demonstrates this key insight: high Property Fidelity does not necessarily translate into successful model execution. Baseline LLMs that appear to generate structurally correct properties often fail at execution time due to missing transitions, unverifiable invariants, or incomplete property sets. SpecMAS trades some property fidelity for more robust executability which is valuable for safety-critical applications requiring provable correctness.
>
> ---
>
> ## 3) Methodology Details
>
> Thank you for this valuable suggestion. Due to page limits, we summarized the methodology and chose to prioritize our core contributions and evaluation in the main paper. We will include a Step-by-Step walkthrough of SpecMAS using a concrete example from MiniSpecBench along with intermediate outputs such as extracted states, transitions, and properties, as well as debugging traces and final verified code in the supplementary section.
>
> ---
>
> ## 4) Benchmark Authenticity
>
> We appreciate this concern and acknowledge that SpecMAS does not use human written SOPs in its evaluation. We opted to use SYNTHIA-generated SOPs because the ground truth NuSMV model files lacked natural language documentation, making aligned SOP and NuSMV model file pairs unavailable. However, SYNTHIA was explicitly designed to generate semantically faithful but syntactically diverse SOPs in natural language from human authored code artifacts. These SOPs paraphrase the system’s functional requirements. We manually checked all generated SOPs to make sure it did not contain any explicit mention of NuSMV style temporal properties or transition logic to avoid trivializing the task. All generated SOPs are available in the supplementary material. However, we acknowledge this limitation and will explore Human-written SOPs in future work.
>
> ---
>
> ## Technical Clarifications
>
> - **Minimal edits (Debugging Steps):** For SpecMAS we recorded the number of steps taken by the debugger agent until a successful execution is reached. For baseline LLMs, it was computed manually by iteratively prompting each LLM model with the corresponding error trace after every execution of its updated NuSMV code. This is done up to a maximum of 60 debug steps.
>
> - SpecMAS is fully automated once initialized and requires no human intervention.
>
> - LLMs do struggle with NuSMV syntax. We encountered cases where LLMs had issues with generating valid NuSMV syntax, inability to follow the set program structure i.e INIT, VAR, ASSIGN etc. Which is precisely why our debugging agent is essential in the architecture of SpecMAS.
>
> We will add a brief limitations section addressing these concerns in supplementary material and expand it with more specifics and edge cases.

---

> ### Comment · Reviewer_YKJx · 2025-08-04
>
> Thank you for your responses. It addresses some of my questions.
>
> I'm still not satisfied with the measurement of specification correctness. The fact that there is a tradeoff between property fidelity and execution compliance is curious. It seems to indicate that property fidelity is not really a rigorous account of whether the generated specifications are actually the correct ones to verify. If it is indeed measuring this, then the message would be that specification synthesis (and verification) is not quite relevant to the quality of the final generated program.

---

> > ### Author Response · Authors · 2025-08-04
> >
> > Property Fidelity, as defined in our paper, measures the logical equivalence and syntactic similarity between the generated properties and the expert-authored reference properties. While this captures alignment with expected formal expressions, it does not guarantee that the generated specifications are verifiable against the corresponding model. A property can be logically sound and semantically well-formed, yet still lead to counterexample traces during model checking due to mismatches in assumptions, missing state conditions, or incorrect variable scoping.
> >
> > This is one of the challenges that we are trying to address with SpecMAS. Our Execution Compliance metric reflects the outcome of formal verification, i.e., whether the properties are actually satisfied by the generated model. A model that scores high on property fidelity but fails during execution is often too strict, incomplete, or incompatible with the synthesized transition system. Conversely, a model with lower property-fidelity score may still be verifiable against all extracted properties, thus achieving a higher Execution Compliance score.

---

> > > ### Comment · Reviewer_5uhY · 2025-08-04
> > >
> > > Thank you. Were there significant challenges in translating from natural language and CTL and LTL?

---

> > > ### Author Response · Authors · 2025-08-07
> > >
> > > following up on our previous comment, we would like to clarify an important distinction about our evaluation metrics.
> > > Your observation about Property Fidelity highlights an important distinction in our evaluation design. Property Fidelity was never intended to measure correctness of specification extraction ,it measures syntactic similarity to expert examples. This is precisely why we don't rely on it as our primary correctness indicator.
> > >
> > > Property Fidelity measures style/syntax similarity, while Execution Compliance measures whether the specifications actually work when formally verified. The trade-off exists because expert-written properties sometimes contain issues that prevent verification.

---

### Note · Authors · 2025-08-13

We thank all reviewers for their thoughtful, constructive, and rigorous feedback. We greatly appreciate the recognition of the novelty and significance of SpecMAS as the first multi-agent framework for generating formally verified system models from natural language SOPs.

In our rebuttal, we have addressed key concerns regarding the correctness and semantic relevance of generated specifications, the rationale for the multi-agent design, the role of property fidelity versus execution compliance, and challenges in translating SOPs to CTL/LTL logic. We have also clarified that SpecMAS generated properties include semantic entailment checks to prevent vacuously satisfied properties, such that debugging decisions are driven by both formal verification outcomes and semantic alignment. We further addressed that we ensure variable consistency among agents through shared intermediate representations built during the Kripke Model Generation phase.

SpecMAS directly tackles a pressing challenge in bridging natural language and formal system verification, and we believe the work offers both foundational contributions and practical tools for future research. We appreciate the valuable dialogue with the reviewers and hope our responses offer sufficient clarity and confidence in the merit of this submission.

---

### Decision · Program_Chairs · 2025-09-17

**Decision:**

Accept (poster)

**Comment:**

This paper introduces SpecMAS, a framework that synthesizes and verifies programs from natural language descriptions, specifically Standard Operating Procedures (SOPs). The approach uses Large Language Models (LLMs) to parse SOPs, generate formal specifications (CTL/LTL), and iteratively correct a finite-state transition system model. This model is then verified using formal methods tools. SpecMAS is shown to outperform LLM-only program synthesis baselines across multiple benchmarks and metrics.

This paper makes a strong contribution by introducing a novel framework that effectively bridges natural language processing and formal methods. It combines LLMs and verification tools, like NuSMV, within a closed-loop architecture to generate programs that provably meet formal specifications extracted from text. The approach is well-executed, with comprehensive experiments demonstrating robustness across multiple domains. While the use of synthetic SOPs somewhat limits real-world applicability and certain implementation details are omitted, the work remains impactful, generalizable, and full of future potential.